

# Long-term Particulate Matter Modeling for Health Effects Studies in California – Part II: Concentrations and Sources of Ultrafine Organic Aerosols

*Jianlin Hu[1]\*, Shantanu Jathar[2], Hongliang Zhang[3], Qi Ying[4], Shu-Hua Chen[5], Christopher D. Cappa[6], and Michael J. Kleeman[6]\**

*[1]Jiangsu Key Laboratory of Atmospheric Environment Monitoring and Pollution Control, Jiangsu Engineering Technology Research Center of Environmental Cleaning Materials, Collaborative Innovation Center of Atmospheric Environment and Equipment Technology, School of Environmental Science and Engineering, Nanjing University of Information Science & Technology, 219 Ningliu Road, Nanjing 210044, China*

*[2]Department of Mechanical Engineering, Colorado State University, Fort Collins CO, USA*

*[3]Department of Civil and Environmental Engineering, Louisiana State University, Baton Rouge LA, USA*

*[4]Zachry Department of Civil Engineering, Texas A&M University, College Station TX, USA*

*[5]Department of Land, Air, and Water Resources, University of California, Davis. One Shields Avenue, Davis, CA, USA*

*[6]Department of Civil and Environmental Engineering, University of California, Davis. One Shields Avenue, Davis CA, USA*

\*Corresponding authors:

Jianlin Hu, Tel.: +86 25 5873 1504; E-mail address: jianlinhu@nuist.edu.cn; hu_jianlin@126.com

Michael J. Kleeman, Tel.: +1 530 752 8386; fax; +1 530 752 7872. E-mail address:

mjkleeman@ucdavis.edu





# 1    Abstract

Organic aerosol (OA) is a major constituent of ultrafine particulate matter ($PM_{0.1}$).
Recent epidemiological studies have identified associations between $PM_{0.1}$ OA and premature
mortality and low birth weight. In this study, the source-oriented UCD/CIT model was used to
simulate the concentrations and sources of primary organic aerosols (POA) and secondary
organic aerosols (SOA) in $PM_{0.1}$ in California for a 9-year (2000 - 2008) modeling period with 4
km horizontal resolution to provide more insights about $PM_{0.1}$ OA for health effects studies. As a
related quality control, predicted monthly average concentrations of fine particulate matter
($PM_{2.5}$) total organic carbon at six major urban sites had mean fractional bias of -0.31 to 0.19 and
mean fractional errors of 0.4 to 0.59. The predicted ratio of $PM_{2.5}$ SOA/OA was lower than
estimates derived from chemical mass balance (CMB) calculations by a factor of 2~3, which
suggests the potential effects of processes such as POA volatility, additional SOA formation
mechanism, and missing sources. OA in $PM_{0.1}$, the focus size fraction of this study, is dominated
by POA. Wood smoke is found to be the single biggest source of $PM_{0.1}$ OA in winter in
California, while meat cooking, mobile emissions (gasoline and diesel engines), and other
anthropogenic sources (mainly solvent usage and waste disposal) are the most important sources
in summer. Biogenic emissions are predicted to be the largest $PM_{0.1}$ SOA source, followed by
mobile sources and other anthropogenic sources, but these rankings are sensitive to the SOA
model used in the calculation. Air pollution control programs aiming to reduce the $PM_{0.1}$ OA
concentrations should consider controlling solvent usage, waste disposal, and mobile emissions
in California, but these findings should be revisited after the latest science is incorporated into
the SOA exposure calculations. The spatial distributions of SOA associated with different
sources are not sensitive to the choice of SOA model, although the absolute amount of SOA can



change significantly. Therefore, the spatial distributions of $PM_{0.1}$ POA and SOA over the 9-year
study period provide useful information for epidemiological studies to further investigate the
associations with health outcomes.
**Key Words:** Primary organic aerosols, secondary organic aerosols, California, sources,
UCD/CIT model.



## 1. Introduction

Organic aerosol (OA) is a significant constituent of fine particulate matter ($PM_{2.5}$) (Zhang
et al., 2007) and a dominant constituent of ultrafine particulate matter ($PM_{0.1}$) (Kleeman et al.,
2009; Sardar et al., 2005a). Epidemiology studies carried out over the past 20 years link $PM_{2.5}$ to
severe short-term and long-term health effects such as asthma, cardio-respiratory disease, and
lung cancer (Dockery, 2001; Dockery and Pope, 1994; Dockery et al., 1993; Franklin et al., 2007;
Le Tertre et al., 2002; Pope et al., 2002; Pope and Dockery, 2006). Epidemiological studies for
$PM_{0.1}$ mass are in the early stages of development but preliminary results show associations with
premature mortality (Ostro et al., 2015) and low birth weight (Laurent et al., 2014). OA is an
important species due to its contribution to $PM_{2.5}$ and $PM_{0.1}$ mass, and the toxicity of some
compounds within OA has motivated even greater scrutiny in health studies (Mauderly and
Chow, 2008). A few $PM_{2.5}$ epidemiology studies have investigated the associations between
exposure to OA and health effects with mixed results (Cao et al., 2012; Krall et al., 2013; Levy et
al., 2012; Mar et al., 2000; Ostro et al., 2006; Ostro et al., 2010). The early epidemiological
studies conducted for $PM_{0.1}$ have identified subcategories of OA that are highly associated with
negative health effects (Laurent et al., 2016a; Laurent et al., 2014; Laurent et al., 2016b; Ostro et
al., 2015) and these results merit further investigation to identify the exact sources and
compound classes that may be related to $PM_{0.1}$ OA toxicity.
The exposure fields used in the published $PM_{0.1}$ epidemiology studies to date have been
generated with chemical transport models (CTMs) because $PM_{0.1}$ measurements with sufficient
spatial or temporal resolution are not widely available. In these studies, predictions using the
UCD/CIT (University of California Davis/California Institute of Technology) model were
evaluated against $PM_{2.5}$ and $PM_{0.1}$ point measurements as a confidence building exercise and the



54 model predictions were then used to estimate exposure fields with ~4km and ~24hr resolution

55 over the state of California (Hu et al., 2014a; Hu et al., 2014b; Hu et al., 2015). The OA exposure

56 fields generated through this approach reflect the state-of-the-science predictions from CTMs at

57 the time they were done, but they may not capture the full complexity of atmospheric OA.  OA

58 consists of primary organic aerosol (POA) and secondary organic aerosol (SOA). POA is directly

59 emitted to the atmosphere in the particle phase and SOA is formed in the atmosphere from the

60 oxidation of volatile or semi-volatile organic compounds (Seinfeld and Pankow, 2003). Both

61 POA and the precursors of SOA can be emitted from anthropogenic and biogenic sources

62 (Mauderly and Chow, 2008). Numerous theories have been put forward about the volatility of

63 POA (Robinson et al., 2007), the conversion of intermediate volatility compounds to SOA

64 (Jathar et al., 2014; Zhao et al., 2014), and the role of water in SOA formation (Jathar et al., 2016;

65 Pankow et al., 2015). A comprehensive model for OA that has been fully constrained by

66 measurements has not been demonstrated to date, which makes it difficult to estimate $PM_{2.5}$ OA

67 exposure using CTMs. However, measurements indicate the OA in the $PM_{0.1}$ size fraction is

68 more heavily influenced by POA (Ham and Kleeman, 2011; Kleeman et al., 2009), which makes

69 estimating exposure to $PM_{0.1}$ using CTMs more feasible.

70  The current paper, as the fourth in the series (Hu et al., 2014a; Hu et al., 2014b; Hu et al.,

71 2015), investigates the UCD/CIT model capability in predicting the concentrations and sources

72 of POA and SOA in $PM_{0.1}$. The objective of this study is to identify the features of the CTM

73 POA and SOA results that could add skill to the exposure assessment for epidemiological studies

74 and to discuss the potential problems in modeling POA and SOA for use in health effects studies.

75



## 2. Methods

### 2.1 Model Description

The source-oriented University of California-Davis/California Institute of Technology (UCD/CIT) air quality model was used to predict OA concentrations in the current study. The UCD/CIT model tracks primary particles and SOA formation from different sources separately through the calculation of all major aerosol processes such as emissions, transport, deposition, gas-to-particle conversion, and coagulation. The standard algorithms of these processes used in the current study are provided in a companion paper (Hu et al., 2015) and references therein, therefore only the details of the algorithms for POA and SOA source apportionment calculation are described here.

The UCD/CIT source-oriented air quality model tracks primary particles emitted from different sources by adding artificial tracers to represent total primary mass contributions from different sources in each particle size bin (Ying et al., 2008). The emissions of tracers are empirically set to be 1% of the total mass of primary particles emitted from each source category, thus the particle radius and the dry deposition rate are not significantly changed. The primary PM total mass concentrations from a given source then are directly correlated with the simulated artificial tracer concentrations from that source. Source specific emission profiles are used to estimate the POA concentrations in the primary PM total mass using the equation (1):

$$POA_{i,j} = C_{i,j} \times A_{i,j} \qquad \qquad (eq.\ 1)$$

where $POA_{i,j}$ and $C_{i,j}$ represent POA concentration and primary PM total mass concentration in size bin $i$ from $j$th source, respectively. $A_{i,j}$ represents OA fraction per unit mass of PM emitted from the $j$th emission source in size bin $i$. More details describing the POA source apportionment





technique and the emission profiles are provided in the previous studies (Ying and Kleeman,
2004; Ying et al., 2008).

The SOA module used in the current study follows the two-product method described by

Carlton et al. (2010). SOA formation is considered from seven precursors: isoprene,
monoterpenes, sesquiterpenes, long-chain alkanes, high-yield aromatics, low-yield aromatics,
and benzene. The seven precursors form twelve semi-volatile products and seven nonvolatile
products. The calculations consider dynamic gas-particle conversion of the semi-volatile and
nonvolatile products. A more detailed description of the SOA module and parameters used in
gas-to-particle transfer calculation is provided in the part I paper (Hu et al., 2015) and references
therein.

The original SOA module described above was modified to have the source

apportionment capability inherent in the UCD/CIT model. SOA source apportionment is
predicted by tracking the SOA precursor emissions from different sources individually through
all atmospheric processes as they react to form low-volatility products that can partition to the
particle phase based on the SOA module described above. This approach was initially developed
for source apportionment of secondary inorganic aerosols, such as nitrate, sulfate, and
ammonium (Mysliwiec and Kleeman, 2002; Ying and Kleeman, 2006). Later, this approach was
applied for SOA source apportionment in California using the Caltech Atmospheric Chemistry
Mechanism (Chen et al., 2010; Kleeman et al., 2007) and in Texas using the SAPRC99
mechanism (Zhang and Ying, 2011; Zhang and Ying, 2012). In the current study, the SAPRC11
mechanism was used and expanded to track the reactions of SOA precursors emitted from
different sources. Chemical reaction products leading to SOA formation are labeled with the



source-identity of the reactant so that source attribution information is preserved. For the
example of benzene (BENZ) reaction with OH forming benzene derived SOA,

BENZ + OH → SV.BNZ1 + SV.BNZ2                          (rx. 1)

SV.BNZ1 ↔ ABNZ1                                        (rx. 2)

SV.BNZ2 ↔ ABNZ2                                        (rx. 3)

where SV.BNZ1 and SV.BNZ2 represents the two semi-volatile products that partition between
gas and particle phase, and ABNZ1 and ABNZ2 represent the particle phase SOA products from
SV.BNZ1 and SV.BNZ2, respectively. If there are two sources for BENZ, then BENZ is
expanded into two species BENZ_X1 and BENZ_X2 in the model. The above pathways (rx1 –
rx3) are then expanded as:

BENZ_X1 + OH → SV.BNZ1_X1 + SV.BNZ2_X1               (rx. 4)

SV.BNZ1_X1 ↔ ABNZ1_X1                                 (rx. 5)

SV.BNZ2_X1 ↔ ABNZ2_X1                                 (rx. 6)

BENZ_X2 + OH → SV.BNZ1_X2 + SV.BNZ2_X2               (rx. 7)

SV.BNZ1_X2 ↔ ABNZ1_X2                                 (rx. 8)

SV.BNZ2_X2 ↔ ABNZ2_X2                                 (rx. 9)

Thus, the SOA products from BENZ ABNZ1_X1, ABNZ1_X2, ABNZ2_X1 and

ABNZ2_X2 contain the information needed to calculate source contributions to the SOA
concentrations.
**2.2 Model Application**

The UCD/CIT model was applied to simulate the concentrations and sources of POA and

SOA during ~ a decadal period (9 years from 2000 January 1$^{st}$ to 2008 December 31$^{st}$) over



California using a one-way nesting technique added to the UCD/CIT model (Zhang and Ying,
2010). The parent domain covers the entire state of California using a 24km horizontal grid
resolution and two nested domains cover the most populated areas (> 92% of California total
population) using a 4km horizontal grid resolution. Emissions of the seven SOA precursors were
grouped into nine source categories: on-road gasoline engines, off-road gasoline engines, on-
road diesel engines, off-road diesel engines, wood smoke, meat cooking, high sulfur fuel
combustion, other anthropogenic sources (including solvent usage, waste disposal emissions etc.),
and the biogenic sources. Primary PM emissions were also grouped into these 9 source
categories. Particulate composition, number and mass concentrations in the range between 0.01
and 10 μm in diameter were represented in 15 size bins with the first 5 bins for $PM_{0.1}$ (0.01 to 0.1
μm) in the model. Biogenic emissions were generated using the U.S. EPA's biogenic emission
inventory system (BEIS3.14). The Weather Research and Forecasting model (WRF) v3.1.1
(William C. Skamarock, June 2008) was used to simulate the 24 km and 4 km hourly
meteorology fields (wind, temperature, humidity, precipitation, radiation, air density, and mixing
layer height) that drove the UCD/CIT model simulations. WRF simulations were initialized and
bounded by the North American Regional Reanalysis (NARR) data with 32 km resolution and 3-
hour time resolution. The four-dimensional data assimilation (FDDA) (Liu et al., 2005)
technique was used and the surface friction velocity (u*) in the WRF model was increased by 50%
to improve the surface wind predictions as suggested by previous studies (Hu et al., 2012; Hu et
al., 2010; Mass, 2010). Details of the modeling domains, vertical cell spacing, preparation of
emissions and meteorological inputs are provided in the first paper in the series (Hu et al., 2015).



## 3. Results

### 3.1 Concentrations of POA and SOA

Hourly POA and SOA concentrations in multiple size fractions were calculated

throughout the 9-year simulation period, and then averaged to daily and monthly average

concentrations. Although the focus of the current study is $PM_{0.1}$ POA and SOA, the predicted

$PM_{2.5}$ OA concentrations were also calculated and compared to measurements as a confidence

building exercise (since $PM_{0.1}$ measurements are not routinely available). Model calculations

predict organic matter (OM) concentrations while ambient measurements quantify organic

carbon (OC) concentrations. Simulated OM concentrations are converted to OC concentrations

using an OM/OC ratio of 1.6 for POA (Turpin and Lim, 2010) and species-specific OM/OC

ratios for SOA species taken from Table 1 in Carlton et al. (2010). Detailed evaluation of the

model performance for $PM_{2.5}$ OC (and other PM / gaseous species) has been presented in the first

paper in the series (Hu et al., 2015). In summary, predicted monthly average $PM_{2.5}$ OC has a

mean fractional bias of -0.32 and a mean fractional error of 0.43. Monthly mean fractional bias

(MFB) and mean fractional errors (MFE) calculated using daily average OC generally meet the

model performance criteria proposed by Boylan and Russell (2006).

Figure 1 illustrates the time series of the predicted and measured monthly-average total

$PM_{2.5}$ OC concentrations at six major urban locations (a) Sacramento, (b) San Jose, (c) Fresno, (d)

Bakersfield, (e) Los Angeles, and (f) Riverside. Measured $PM_{2.5}$ OC concentrations at all sites

show strong seasonal variation with higher concentrations in winter months and lower

concentrations in summer months. OC concentrations predicted by the UCD/CIT model

generally capture the monthly average concentrations and seasonal variations with MFB ranging



from -0.31 to 0.19 and MFE ranging from 0.4 to 0.59. However, the model predicts much weaker
trends of $PM_{2.5}$ OC over the 9 years at Los Angeles and Riverside, indicating that the declining
emission trends might not be well represented in the inventory. At Sacramento and Fresno, the
measured monthly average OC concentrations frequently exceeded 10 µg/m$^3$ in winter and the
maximum monthly OC concentrations reached or exceeded ~25 µg/m$^3$. Wood smoke is predicted
to be the dominant OC source at the two locations, contributing over 70% of the total OC
concentrations on average. Wood smoke is also predicted to be the dominant OC source in
winter at San Jose and Bakersfield. Model calculations tend to over-predict the winter OC
concentrations at San Jose, indicating the wood smoke emissions are likely over-estimated in this
area. Model calculations generally under-predict OC in summer when concentrations are lower.
Meat cooking and other anthropogenic sources are predicted to be the largest sources in summer
at Sacramento, San Jose, Fresno, and Bakersfield. Together these two categories contribute over
86% of the total predicted OC in summer. Both measured and predicted seasonal variation is
weaker at Los Angeles and Riverside than in Northern California due to smaller wood smoke
contributions. Meat cooking and other anthropogenic sources make the largest predicted
contributions to OA at these two Southern California locations. Mobile sources (gasoline and
diesel engines) also contribute approximately 30% of the total $PM_{2.5}$ OC at Los Angeles. Model
calculations tend to under-predict $PM_{2.5}$ OC concentrations in all seasons in 2000-2006 at
Riverside (approximately 80 km downwind of the Los Angeles urban center). Intense emissions
transported from the upwind Los Angeles areas along with the meteorology and topography
enhances photo-oxidation of volatile organic compounds (VOCs) and formation of SOA at this
location. A measurement study of organic aerosols at Riverside in summer indicated high SOA
fraction of the total OA with an average SOA/OA ratio of 0.74 (Docherty et al., 2008). The





PM$_{2.5}$ OC under-prediction at Riverside during summer and the general under-prediction in
summer at other sites may indicate that some important precursors and pathways of PM$_{2.5}$ SOA
are missing or only partially included in the current SOA module, such as SOA formation from
glyoxal and methylglyoxal (Ervens and Volkamer, 2010; Fu et al., 2008; Ying et al., 2015) and
from aerosol aqueous phase chemistry (Volkamer et al., 2009), the conversion of intermediate
volatility compounds to SOA (Jathar et al., 2014; Zhao et al., 2014), or SOA forming with higher
yields than included in the module (Zhang et al., 2014; Cappa et al., 2016).

Figure 2a compares the average PM$_{2.5}$ OC/mass ratios estimated from ambient

measurements and the values predicted by the UCD/CIT model over the 9-year study period at
seven representative urban locations. At each site, daily average measured concentrations of the
PM$_{2.5}$ total mass and OC were obtained from California Air Resources Board (CARB) (CARB,
2011) "1 in 3" sampling network and averaged over the 9 year period. Predicted concentrations
on the corresponding days were extracted and averaged for the comparison.  The average
OC/mass ratios were then calculated. The observed average OC/mass ratios vary in the range of
0.24 (at Riverside) to 0.45 (at Sacramento). The predicted average OC/mass ratios are in
relatively good agreement with measured values at Los Angeles, Riverside, and Bakersfield
(difference < 20%), but not at Sacramento, San Jose, Fresno, and El Cajon (difference > 35%).
The predicted average OC/mass ratios are consistently lower than observed ratios, by 0.01 (3% at
Los Angeles) to 0.22 (48% at Sacramento). This under-prediction is partly attributed to the
under-prediction of OC concentrations, especially the SOA concentrations, but also to the over-
prediction of total mass concentrations due to over-estimated dust emissions (Hu et al., 2014a;
Hu et al., 2015). A sensitivity analysis was conducted by removing the dust concentrations from
the predicted PM$_{2.5}$ mass (Figure 2a). The average predicted OC/mass ratio increased from 0.22





to 0.29 (average across the seven sites), compared to the observed ratio of 0.33. Omission of dust
from the model predictions improves agreement with OC/mass measurements at all sites except
central Los Angeles, although OC/mass without dust is still lower than measurements at four
sites (Sacramento, San Jose, Fresno, and El Cajon) indicating OC predictions are likely biased
low at these locations.

Figure 2b compares the predicted and observed OC/mass ratios in the ultrafine ($PM_{0.1}$) or

quasi-ultrafine ($PM_{0.18}$, $PM_{0.25}$) particles. The ultrafine/quasi-ultrafine measurement data were
compiled in a previous study (Hu et al., 2014a) from published literature (Herner et al., 2005;
Kim et al., 2002; Krudysz et al., 2008; Sardar et al., 2005a; Sardar et al., 2005b). The ultrafine or
quasi-ultrafine data are more sparse than the $PM_{2.5}$ data, but still cover a sufficient total number
of days to allow for robust comparison. The observed OC/mass ratios in ultrafine/quasi-ultrafine
sizes vary from 0.43 (at Modesto) to 0.71 (at USC). The predicted ultrafine/quasi-ultrafine
OC/mass ratios generally agree well with observed values at all sites. The generally better
agreement of OC/mass ratios in the ultrafine/quasi-ultrafine size range compared to the $PM_{2.5}$
size range reflects the fact that SOA formation and dust emissions make limited contributions to
ultrafine/quasi-ultrafine concentrations. Condensation of SOA mostly takes place in the particle
accumulation mode, and is generally not dominant in the ultrafine size range due to the increase
in the saturation vapor pressure above small particles (Kelvin effect). Dust components mainly
contribute to coarse and fine particles, but make little contribution to the ultrafine particles.

The primary and secondary fraction of total OA cannot be directly measured in ambient

OA samples. A few indirect methods have been developed to estimate the POA and SOA
concentrations, such as molecular marker-based method (Daher et al., 2011; Daher et al., 2012;
Ham and Kleeman, 2011; Kleindienst et al., 2007), elemental carbon (EC) tracer method





(Cabada et al., 2004; Lim et al., 2003; Polidori et al., 2007; Polidori et al., 2006; Turpin and
Huntzicker, 1995), water soluble organic carbon content method (Weber et al., 2007), aerosol
mass spectrometry factorization method (Aiken et al., 2008; Lanz et al., 2007; Ulbrich et al.,
2009), and the un-explained fraction of OA by tracers for major POA categories (Chen et al.,
2010; Schauer and Cass, 2000). In the current study, $PM_{2.5}$ SOA concentrations were estimated
by the molecular marker Chemical Mass Balance (CMB) method  (Daher et al., 2012) during
sampling periods in 2005-2007 at four locations. $PM_{2.5}$ POA concentrations were then estimated
by subtracting $PM_{2.5}$ SOA concentrations estimated by the CMB method from the total measured
OA concentrations. Figure 3 shows the $PM_{2.5}$ POA and SOA concentrations predicted by the
UCD/CIT model (right dark columns) compared to the $PM_{2.5}$ POA and SOA concentrations
estimated using the CMB method (left gray columns). Error bars represent the standard deviation
of concentrations estimated during the sampling periods. The total $PM_{2.5}$ OA (i.e., POA + SOA)
concentrations predicted by the UCD/CIT model generally agree with measured values (with
fractional bias within ±35%) except at the Riverside site (with a fraction bias of -63%). But the
$PM_{2.5}$ SOA concentrations predicted by the UCD/CIT model appear to be a factor of 2~3 lower
than the SOA concentrations estimated by the CMB method (ratio ranging from 2.2 at Riverside
to 2.8 at WSanG). The $PM_{2.5}$ POA concentrations predicted by the UCD/CIT model are higher
than those estimated by the CMB method at WSanG and ESanG1.  This may reflect the effects
of POA volatility. Studies have indicated that some fraction of POA emissions will evaporate,
and this material may undergo photo-oxidation and condense back to particle phase (Robinson et
al., 2007). In the current model, POA is treated as non-volatile. Thus, no such evaporation occurs.
However, the substantial under-prediction of $PM_{2.5}$ SOA at all sites suggests that some SOA
precursors and pathways are likely missing from the current SOA mechanism. Both $PM_{2.5}$ POA



and SOA are under-predicted at Riverside, indicating that some important sources are likely
missing in that area.

Figure 4 illustrates the predicted total $PM_{0.1}$ OA concentrations (Figure 4a) and the

predicted ratios of SOA to total OA averaged over the 9 year modeling period (Figure 4b). High
total $PM_{0.1}$ OA concentrations with maximum concentrations $> 2$ µg/m$^3$ are located in urban
areas where the POA emissions are large due to human activities. Predicted $PM_{0.1}$ SOA generally
accounts for less than 10% of total $PM_{2.5}$ OA at urban areas, but predicted SOA contribute to
10~20% of total OA in suburban areas, and contribute to 20~50% in rural areas. The spatial
distribution of $PM_{2.5}$ SOA concentrations and the SOA to total OA ratios (shown in Figure S1)
are generally similar to those of $PM_{0.1}$, but $PM_{0.1}$ OA has sharper spatial gradients and the $PM_{0.1}$
SOA fraction is lower than $PM_{2.5}$, indicating POA contributes more in the ultrafine size range.

Figure 5 shows the contributions from the 9 precursor species to the $PM_{0.1}$ SOA

concentrations (results of $PM_{2.5}$ SOA are shown in Figure S2). Maximum SOA concentrations
are located in southern part of the SJV. Monoterpenes, sesquiterpenes, oligomers, and long
alkanes are the most important precursors, contributing over 90% of the total SOA in most areas,
while other precursors (xylene, toluene, and benzene) in total contribute less than 10 ng/m$^3$ to
SOA concentrations.  These finding are very dependent on the treatment of vapor wall losses
during the formulation of the SOA model. The contributions from different precursors to SOA
concentrations have very different spatial distributions. Long chain alkanes form SOA mainly in
the urban areas of Southern California and in the middle-southern portion of the SJV. Isoprene,
monoterpenes, and sesquiterpenes form SOA at coastal and foothill locations where the biogenic
emissions are greatest. The longer lifetime of long chain alkanes than isoprene leads to a broader
spatial distribution for the SOA derived from alkanes. The spatial distribution of oligomers of





anthropogenic SOA (Oligomer_A) and biogenic SOA (Oligomer_B) reflects the patterns of SOA
derived from long chain alkanes and the total biogenic species.  The relative spatial patterns
associated with each precursor are generally not sensitive to the exact formulation of the SOA
model (see section 3.3).
**3.2 Sources of POA and SOA**
Figure 6 displays the time series of monthly average $PM_{0.1}$ SOA source contributions at
the six major urban locations. $PM_{0.1}$ SOA concentrations are high in summer (100~300 $ng/m^3$)
and low (20~50 $ng/m^3$) in winter, reflecting the seasonal variation in photochemistry. $PM_{0.1}$ SOA
concentrations are higher at Fresno and Bakersfield than other sites due to larger biogenic source
contributions. Biogenic emissions are the largest source of $PM_{0.1}$ SOA across all sites, followed
by the other anthropogenic sources (mainly solvent usage and waste disposal emissions, see
Figure S5). On-road gasoline engines are an important source of SOA at Los Angeles and
Riverside. Similar source contributions to $PM_{2.5}$ SOA are found and shown in Figure S3 in the
Supplemental Materials.
Figure 7 shows the predicted regional source contributions of $PM_{0.1}$ POA averaged over
the 9 year modeling period. The dominant regional sources of $PM_{0.1}$ POA are predicted to be
wood smoke, meat cooking, other anthropogenic sources, on-road gasoline and off-road diesel.
Wood smoke is the dominant POA source especially in Northern California, with the maximum
$PM_{0.1}$ POA contribution exceeding 1 $\mu g/m^3$. Meat cooking and mobile (on-road and off-road)
sources are the major sources in urban areas, especially in metropolitan areas such as Greater Los
Angeles Area and the San Francisco Bay Area. Other anthropogenic sources is another major
category in the urban centers in the SJV and also the Los Angeles areas. High sulfur content fuel



sources are mainly located around the ports in the Los Angeles and San Francisco Bay areas. The
regional source contributions of $PM_{0.1}$ POA are quite different from those of $PM_{2.5}$ POA (shown
in Figure S4). The $PM_{2.5}$ POA source contributions are much more widespread than the $PM_{0.1}$
POA sources contributions because $PM_{2.5}$ has a longer lifetime due to slower deposition and
coagulation compared to $PM_{0.1}$. For example, the mobile sources and the other anthropogenic
sources contribute greatly to $PM_{2.5}$ POA throughout the entire SJV, but only contribute to $PM_{0.1}$
POA in urban centers.

Figure 8 shows the predicted regional source contributions of $PM_{0.1}$ SOA averaged over

the 9 year modeling period (and Figure S6 shows the $PM_{2.5}$ SOA results). Biogenic emission is
predicted to be the single largest $PM_{0.1}$ SOA source in the present study. The maximum biogenic
$PM_{0.1}$ SOA concentration is up to 0.1 $\mu g/m^3$ around Bakersfield in the southern SJV. Other
anthropogenic sources, on-road gasoline engines, and off-road gasoline engines are predicted to
be the dominant anthropogenic sources of $PM_{0.1}$ SOA in California. The spatial distribution of
$PM_{0.1}$ SOA concentrations from these anthropogenic sources are similar (but different from the
spatial distribution of SOA from biogenic sources) with high concentrations in Southern
California. $PM_{0.1}$ SOA formation from on-road diesel engines, off-road diesel engines, wood
smoke, meat cooking and high sulfur fuel combustion are small, with $PM_{0.1}$ SOA contributions
generally less than a few $ng/m^3$. A recent epidemiological study has revealed that anthropogenic
$PM_{0.1}$ SOA is highly associated with ischemic heart disease mortality (Ostro et al., 2015).
Therefore, the results in this study suggest that control of solvent usage, waste disposal, and
mobile emissions should be considered to protect public health in California, but the exact
determination of source controls will need to be evaluated after the SOA formation mechanism is
updated.





### 3.3 Influence of vapor wall losses on SOA exposure in California


The SOA concentrations predicted in the current study are based on the SOA yield data

measured in chamber experiments. A recent study has demonstrated that organic vapors can be
lost to chamber walls during SOA formation experiments resulting in SOA yields that are biased
low (Zhang et al., 2014). Efforts have been carried out to parameterize the effect of vapor wall
losses on SOA formation in the UCD/CIT air quality model to account for this effect when
predicting ambient SOA concentrations in Southern California (Cappa et al., 2015). SOA
concentrations are predicted to increases by factors of 2-5 with low vapor wall loss rates, and by
factors of 5-10 with high vapor wall loss rates, compared to the concentrations in the simulations
with no consideration of vapor wall losses. Here we further analyzed the changes in the
population weighted concentrations (PWCs) of SOA when vapor wall losses are accounted for.
Two sets of simulations (scenarios) conducted by Cappa et al (2015) are considered, one with the
low-$NO_x$, high-yield parameters (denoted as "highyield") and the other with high-$NO_x$, low-yield
parameters (denoted as "lowyield"). Each set of simulations included three vapor wall loss cases,
i.e., no consideration of vapor wall losses (denoted as "base"), low vapor wall loss rates (denoted
as "lowwallloss"), and high vapor wall loss rates (denoted as "highwallloss"). PWCs of SOA are
calculated for six counties in the Southern California for the six scenarios, respectively. Spatial
difference in exposure is important in cohort studies, therefore the relative changes of PWCs
among counties are examined. Figure 9 shows the PWCs of SOA and their relative changes in
different scenarios in the six counties. The results indicate that PWCs of SOA increase
substantially by accounting for vapor wall losses in all counties (panel a). However, the spatial
pattern of SOA PWC, as characterized by normalizing the PWC for each location by the PWC in
Orange County, is very similar in all scenarios (panel b). Consequently, accounting for vapor



wall losses changes the SOA exposure ratio in different counties by only a small extent of < 15%
for most scenarios/counties (panel c). These results suggest that future simulations that account
for vapor wall losses in SOA simulations will yield increased absolute values of concentrations
but will have spatial patterns that are similar to the basecase results in the current paper when
used for epidemiology studies.
Figure 9 suggests that associations between anthropogenic SOA and health effects identified in
previous epidemiological studies will prove robust to future updates in SOA models.  This
finding also extends to the spatial pattern of individual SOA precursors. The influence of vapor
wall losses on exposure to SOA formed from different precursors (i.e., long alkanes, aromatics,
isoprene, sesquiterpenes, and monoterpenes) is shown in Figures S7-S11. In all cases, the spatial
pattern of PWC for SOA derived from each precursor is similar under all treatments of wall
losses.  Long alkanes and aromatics are mainly from anthropogenic sources, and isoprene,
sesquiterpenes, and monoterpenes are mostly from biogenic sources. Further detailed
interpretation of source contributions to SOA and associated health effects should only be carried
out after new exposure fields are calculated using the latest SOA models.

## 4. Conclusions

The source-oriented UCD/CIT model was applied to predict the concentrations and

sources of $PM_{0.1}$ POA and SOA in California for a 9 year (2000 - 2008) modeling period with 4
km horizontal resolution to provide data for health effects studies. As a confidence building
measure, predicted total $PM_{2.5}$ OC concentrations (primary + secondary) and the $PM_{2.5}$ and $PM_{0.1}$
OC/mass ratios generally agree with measured values at fixed point locations. Compared to the
POA and SOA concentrations estimated from measurements at 4 sites using the CMB method,





the $PM_{2.5}$ total OA concentrations predicted by the UCD/CIT model have a fractional bias within
±35% except at the Riverside site. The CMB model estimated $PM_{2.5}$ SOA concentrations
accounted for 13-37% of total OA while the UCD/CIT SOA concentrations accounted for 4-11%
of total OA.  POA volatility, incomplete SOA formation mechanism, and/or missing sources may
account for the discrepancy. For these reasons, the current study focuses on the $PM_{0.1}$ size
fraction.

$PM_{0.1}$ OA has larger contributions from primary sources than the $PM_{2.5}$ size fraction.

Wood smoke is found to be the single biggest source of $PM_{0.1}$ OA in winter in California, and
meat cooking, mobile sources and other anthropogenic sources (mainly solvent usage, and waste
disposal) are the most important sources in summer, but these rankings are sensitive to the SOA
model used in the calculation. Biogenic emissions are predicted to be the largest $PM_{0.1}$ SOA
source, followed by the other anthropogenic sources, and mobile sources. A recent
epidemiological study has revealed that anthropogenic $PM_{0.1}$ SOA is highly associated with
ischemic heart disease mortality (Ostro et al., 2015). Therefore, the results in the present study
suggest that control of solvent usage, waste disposal, and mobile emissions should be considered
to protect public health in California, but detailed source control programs can only be carried
out after revised calculations are performed using updated SOA models. The predicted spatial
distributions of the concentrations and sources of POA and SOA in $PM_{0.1}$ over the 9-year periods
provide detailed information for epidemiological studies to further investigate the associations
with other health outcomes, and these spatial patterns are generally not sensitive to the treatment
of wall losses in the SOA model formulation. All model results included in the current
manuscript can be downloaded free of charge at http://faculty.engineering.ucdavis.edu/kleeman/.





**Acknowledgement**
This study was funded by the United States Environmental Protection Agency under Grant No.
R83386401. Although the research described in the article has been funded by the United States
Environmental Protection Agency it has not been subject to the Agency's required peer and
policy review and therefore does not necessarily reflect the reviews of the Agency and no official
endorsement should be inferred.

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



Figures and Tables

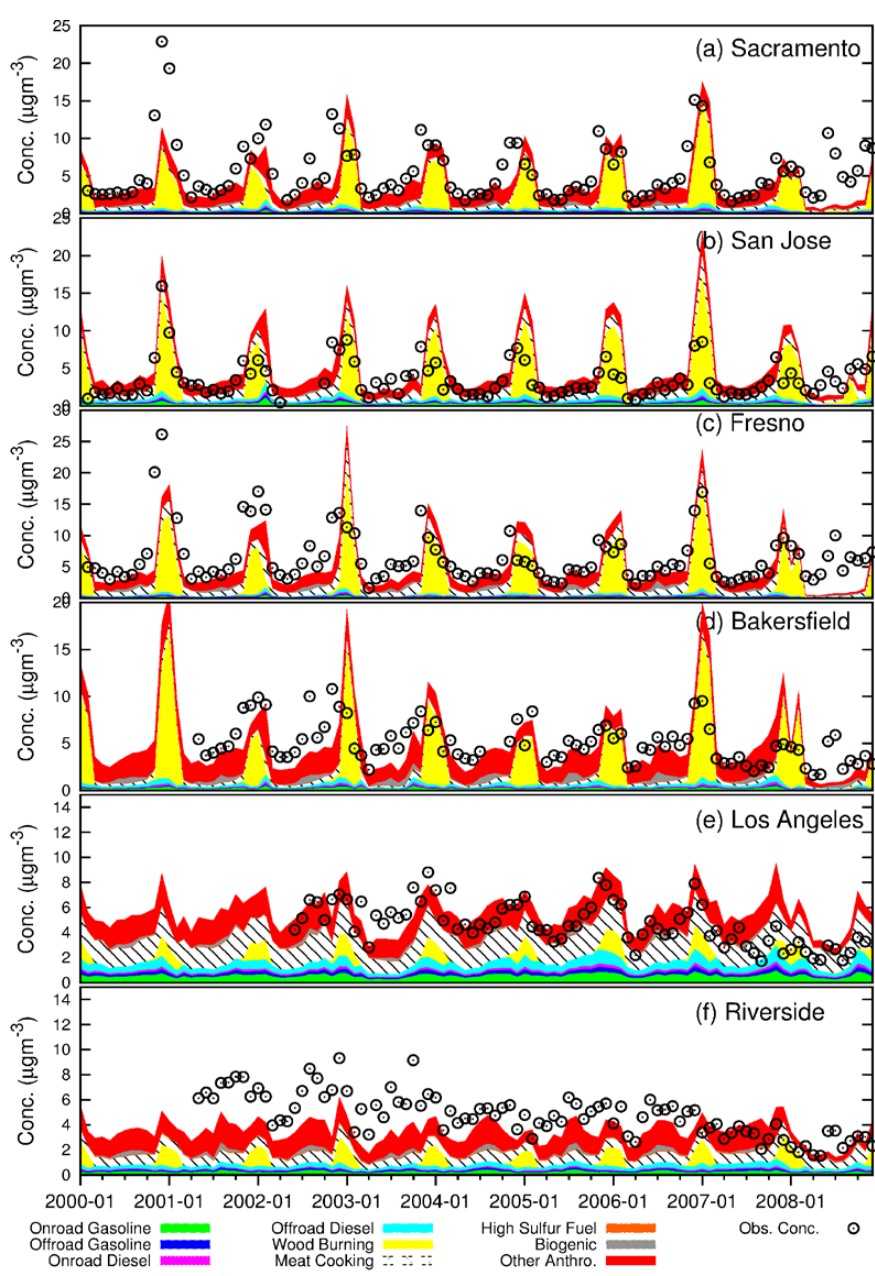


Figure 1. Monthly source contributions to PM$_{2.5}$ total OC at 6 urban sites. Observed total OC
concentrations are indicated by the dot-circles, and predicted OC concentrations from different
sources are indicated by the colored areas.



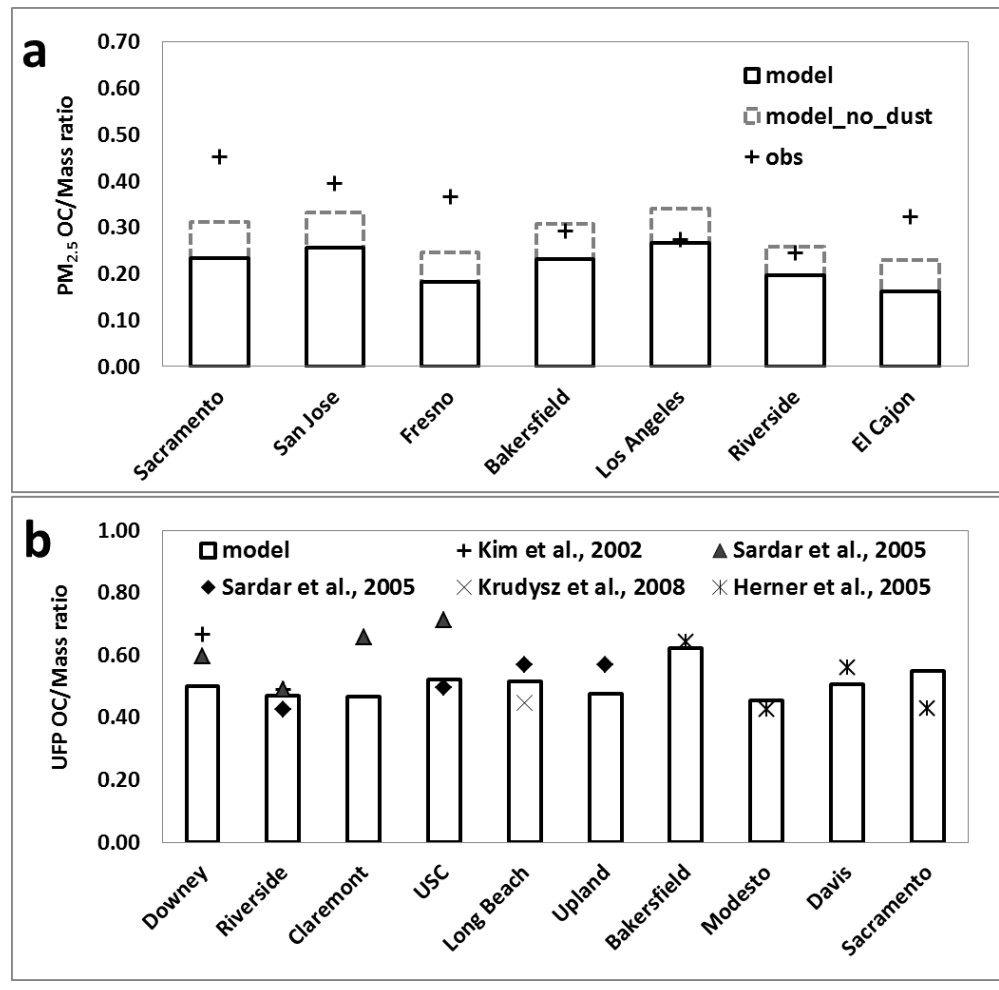


Figure 2. Observed (obs) and predicted (model) OC/Mass ratios in (a) PM$_{2.5}$ and (b) ultrafine and
quasi-ultrafine PM. In (a), a sensitivity analysis is conducted by removing the dust concentration
from the PM$_{2.5}$ total mass (model_no_dust).The ultrafine and quasi-ultrafine data in (b) are
extracted from published literature as indicated in the figure.






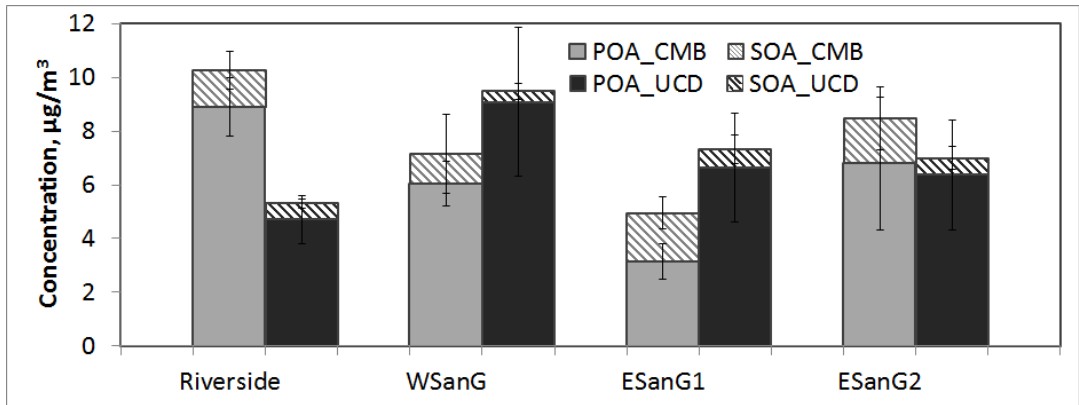


Figure 3. POA and SOA concentrations estimated by the CMB method (left gray columns) and
predicted by the UCD/CIT model (right dark columns). Error bars represent the standard
deviation of concentrations estimated during the sampling periods by both methods. The data are
for sampling periods in 2005-2007 at four sites in Southern California.





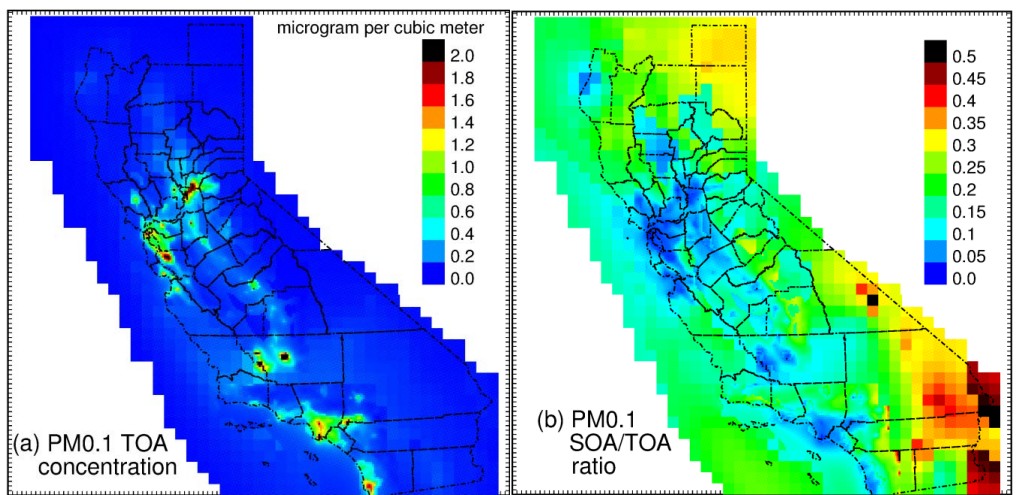

Figure 4. Predicted 9-year average (a) $PM_{0.1}$ Total OA (TOA) concentration and (b) $PM_{0.1}$ SOA/TOA ratio in California.



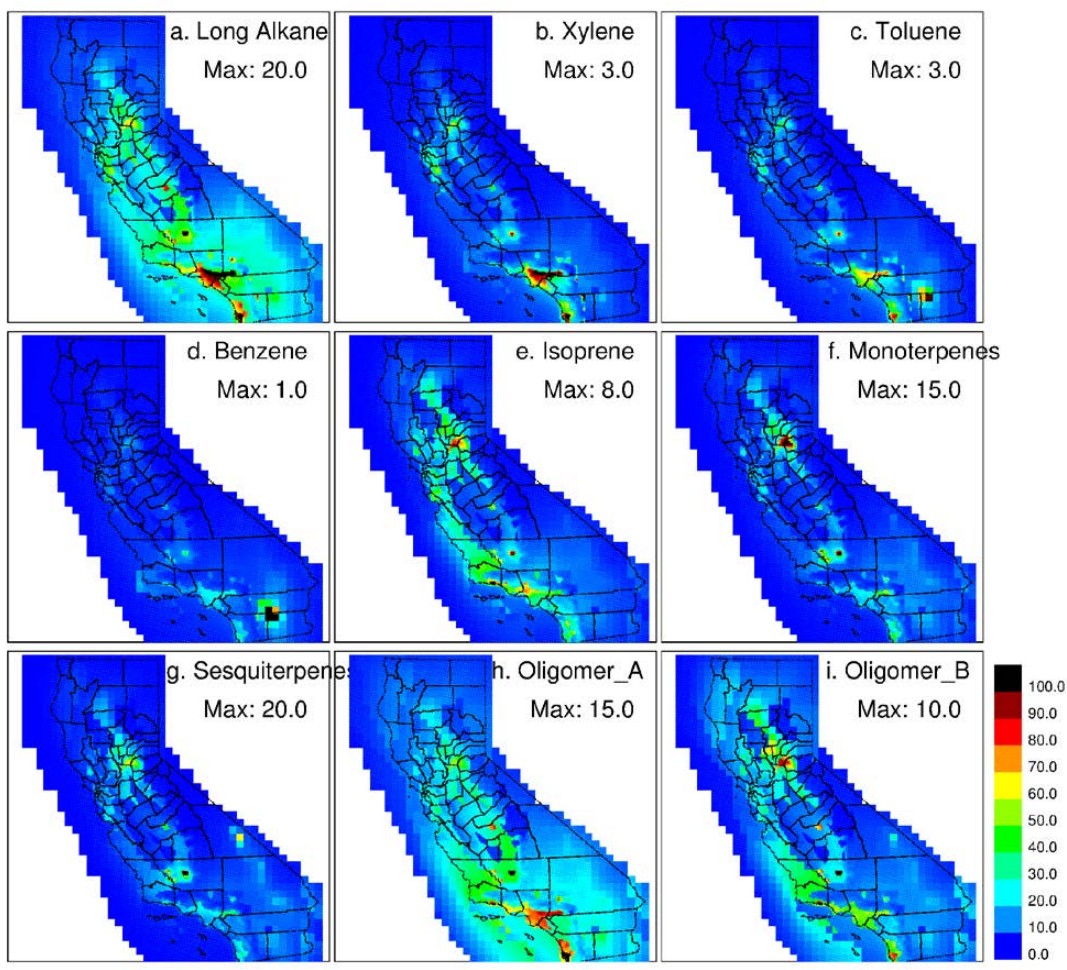

667

Figure 5. The 9-year average $PM_{0.1}$ SOA concentrations derived from (a) AALK, b) AXYL, c) ATOL, d) ABNZ, e) AISO, f) ATRP, g) ASQT, h) AOLGA, and i) AOLGB. Note AXYL and ATOL are actually derived from lumped aromatics species ARO2 (groups of aromatics with $kOH > 2 \times 10^4$ $ppm^{-1}$ $min^{-1}$, including xylenes and other di- and polyalkylbenzenes) and ARO1 (groups of aromatics with $kOH < \times 10^4$ $ppm^{-1}$ $min^{-1}$, including toluene and monoalkylbenzenes). The color scales (shown in the last panel in unit of %) indicate the ratios of the concentrations to the maximum values, which are shown in the panels under species names with a unit of $ng/m^3$.







Figure 6. Monthly source contributions to PM$_{0.1}$ SOA at 6 urban sites. Predicted PM$_{0.1}$ SOA
concentrations from different sources are indicated by the colored areas.





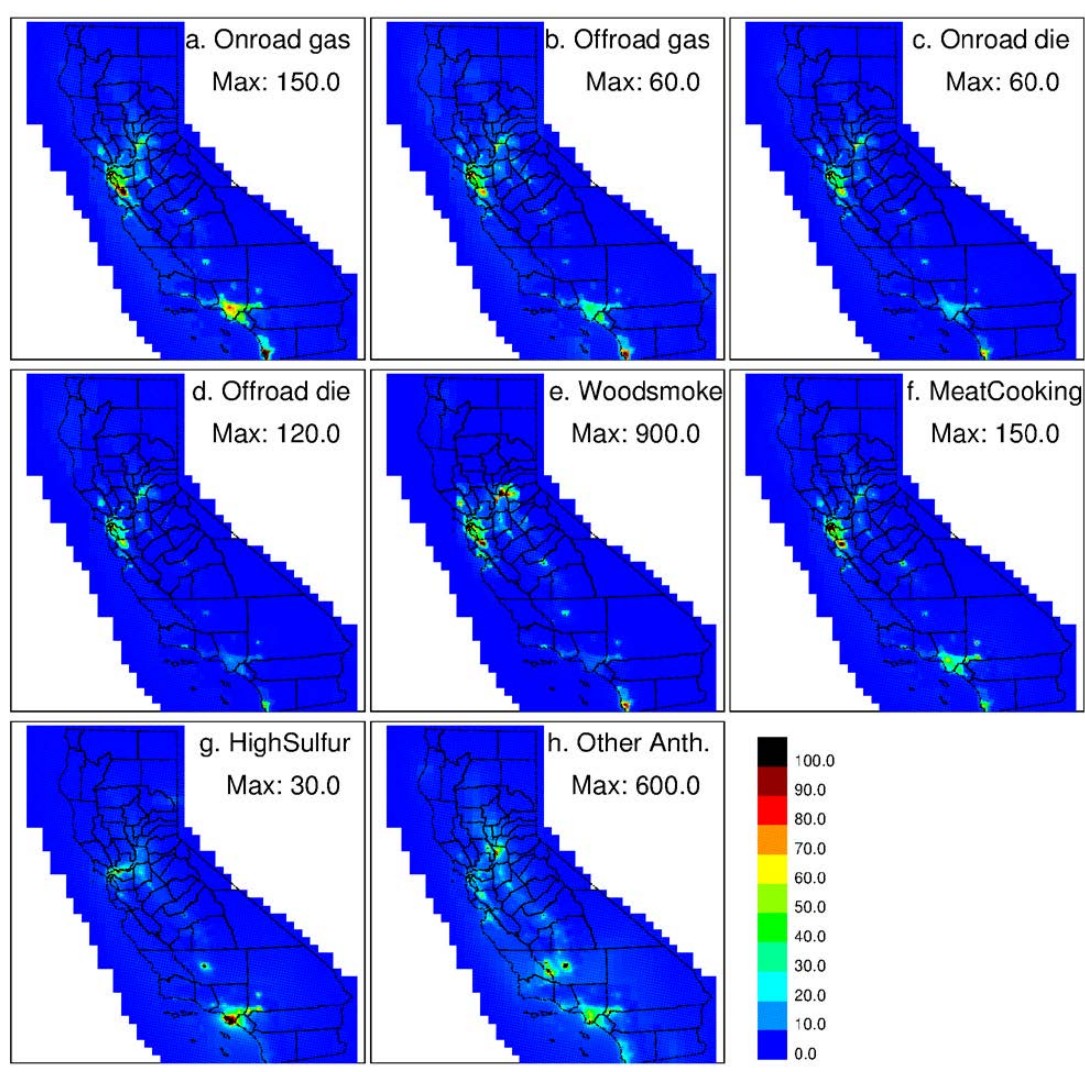


Figure 7. Predicted source contributions to 9-year average PM$_{0.1}$ POA concentrations. The color
scales (shown in the last panel in unit of %) indicate the ratio of the concentrations to the
maximum concentration values, which are shown in the panels under source names with a unit of
ng/m$^3$.





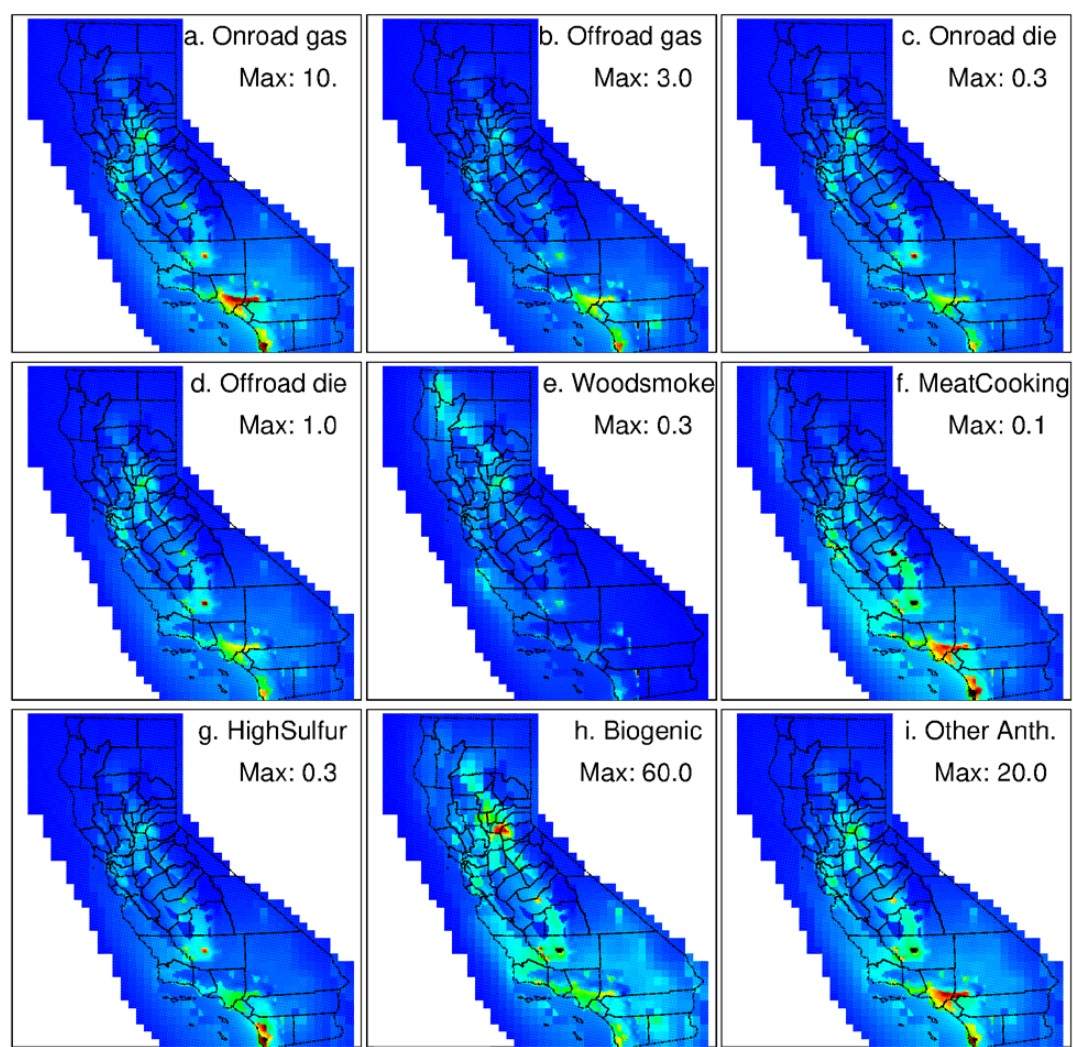


Figure 8. Predicted source contributions to 9-year average $PM_{0.1}$ SOA concentrations. The
definition of the color scales are the same as in Figure 7.










Figure 9. (a) Predicted population weighted concentrations (PWCs) of SOA in six counties in
Southern California. Two sets of simulations (scenarios) conducted by Cappa et al (2015) were
used, one with the low-$NO_x$, high-yield parameters (denoted as "highyield") and the other with
high-$NO_x$, low-yield parameters (denoted as "lowyield"), and each set of simulations included
three vapor wall loss cases, i.e., no considering of vapor wall losses (denoted as "base"), low
vapor wall loss rates (denoted as "lowwallloss"), and high vapor wall loss rates (denoted as
"highwallloss"). (b) Normalized PWCs of SOA in all counties to the PWC of SOA in Orange
County. (c) Changes in the normalized PWCs of SOA in all counties by accounting for vapor
wall losses.