# Peer review of "Long-term Particulate Matter Modeling for Health Effects Studies"

_Atmospheric Chemistry and Physics, 2016_

## Referee Comment (RC1) · Anonymous Referee #3 · 27 Dec 2016

General Comments: This manuscript presents results of concentrations and sources Ultrafine Organic Aerosols from 9 years chemical transport model simulations, which is important for health effects studies. The presentation quality is excellent and no major scientific problems with the presentation. I only have some minor concerns, which are listed below.

Specific comments:

1. In model evaluation, the authors claim that their calculated monthly MFB and MFE meet Boylan and Russel, (2006) standard. It is better to point out the standard in manuscript.

[Figure]

2. Emission inventory is a key part in air quality modeling, but the authors do not write much about emission inventory (based on which year, what the sectors are, and etc.). Besides, the simulation period is as long as 9 years. It is better if the authors can consider the changes in emissions over this period.

3. The authors compared ratio from model to CMB derived ratio. Are the CMB derived ratios 100% reliable? How is its uncertainty? And it is also better to introduce the CMB method in the manuscript.

4. The authors discuss the impact of vapor wall losses on SOA concentrations. How will it affect model evaluation? In Figure 1, model overpredicts OA at some sites. After vapr wall losses correction, model may mismatch with observations.

---

## Referee Comment (RC2) · Anonymous Referee #2 · 31 Dec 2016

The authors provided a long-term analysis for the spatial distribution of PM0.1, and its components including POA and SOA. By using the source apportionment method, the authors further discussed the contribution of different sources on PM0.1 and its components. The article is generally well-written, and has clearly expressed the conclusions clearly by showing convincing data analysis. I will suggest the paper published on ACP, after the authors address my following suggestions:

Pg 10: define the metric used for the evaluation: MFB and MFE. Can put the equation into the Supporting.

Pg 10: "in the first paper in the series": if the authors claimed this paper as "the fourth in the series" (Pg 5 line 70), then I suggest the authors change "the first paper" to "the

third . . ..” to avoid confusion, or do the other way.

Pg 11, line 189-190: I assume the authors were still talking about the winter when they say "Wood smoke is predicted to be . . ..”?

Pg 11, line 192-193: the authors implied that the overestimation of the PM2.5 in the San Jose site was due to the overestimated emission inventory. So how did the authors make that conclusion? Was the emission inventory data significant different from the other places, or more uncertain compared with others?

Pg 12: line 217-219: I suggest the authors move the brief introduction of the 6 Obs sites into Pg 10 to Pg 10 in front of Fig. 1. Also can the authors comment why they didn't use the El Cajon site to evaluate the model's performance of simulating in PM2.5 in Fig. 1?

Pg 15, line 287: change "PM2.5" to "PM2.5-SOA fraction" or "that in PM2.5". Also the authors concluded that the SOA fraction in PM0.1 lower than that in PM2.5, but in Figure 4, we can see the fractions are higher in PM0.1 than PM2.5 in rural areas. Can the authors explain why?

Pg 34, in Figure 7 and others, also in the supplementary, I am confused about the meaning of colorbar. I thought it stands for the fractions from each source category in the total PM0.1 POA, but it seems not. What is the "maximum concentration value", maximum of the monthly mean or maximum of the yearly mean? Also how the authors made the conclusion that the dominant regional sources are "wood smoke, meat cooking . . ."? Looking at the map, most of the data are in the range of "0-10" %, and you can't tell which regions are in the 1% and which regions are in the 9%. For sources with a Max value of 900 but fractions around 1% may not be larger than the source with a Max value of 120 and fractions around 9%. Please quantify the fractions from each source before making conclusion. Also consider doing this for other similar plots.

Pg 43 & 44: Switch the order S4 and S5 to follow when they are mentioned in the

paper.

---

## Referee Comment (RC3) · Anonymous Referee #1 · 3 Jan 2017

The authors discussed the concentrations and sources of primary and secondary organic aerosols in PM0.1 over California for 2000-2008 using the source-oriented UCD/CIT model. The article is overall well-written. I will suggest the paper accepted by ACP after the authors address my following questions/suggestions:

1. SOA module

The SOA module used in this study is based on the two-product method. Different SOA formation treatments could result in different results. It would be meaningful if an alternate SOA module (e.g., VBS) is applied in the future study of POA and SOA.

2. OC/Mass ratios

[Figure]

The authors discussed the underpredictions of OC/Mass ratios shown in Figure 2, which could be due to the overestimation of dust emissions. Are dust emissions affected by wind speed from WRF? Did the authors evaluate the meteorology provided by WRF? What about seasalt, since some sites are along the coast? Although most of seasalt are coarse particles, they may contribute a little bit to PM2.5?

Some other comments:

1. Page 6, line 78, UCD/CIT has been defined in page 4 line 52

2. Page 8, line 136-137, do you mean BENZ (i.e., ABNZ1_X1, ABNZ1_X2, ABNZ2_X1, and ABNZ2_X2)?

3. Page 9, line 155, change "meteorology fields" to "meteorological fields"

4. Page 13, line 246, "Condensation of SOA", do you mean the condensation of volatile VOCs?

5. Page 15, line 277, "some important sources", could you please provide some specific sources that for Riverside case?

6. Page 15, line 287, do you mean less POA converted to SOA in ultrafine size range?

7. Page 16, line 311, You may want to switch the order of supplementary figures.

8. Page 31, Figure 4b, why are PM0.1 SOA/TOA ratios very high over the southeastern corner? Is that partly due to boundary conditions?
* * *

---

## Author Comment (AC1) · 15 Mar 2017

Dear Reviewer,

Thank you for the comments to help improve the quality of the paper. We have revised the manuscript to address your comments and a detailed response to each comment is provided in this file.

The comments are in regular font, the responses are in red, and the changes in the manuscript are in blue.

**Anonymous Referee #3**

General Comments: This manuscript presents results of concentrations and sources Ultrafine Organic Aerosols from 9 years chemical transport model simulations, which is important for health effects studies. The presentation quality is excellent and no major scientific problems with the presentation. I only have some minor concerns, which are listed below.

Specific comments:

1. In model evaluation, the authors claim that their calculated monthly MFB and MFE meet Boylan and Russel (2006) standard. It is better to point out the standard in manuscript.

Responses: The PM model performance criteria of MFB and MFE, suggested by Boylan and Russell (2006), are a function of PM concentration as follows,

$$MFB\ (\%) \leqslant \pm 140e^{-(Co+Cm)} + 60$$

$$MFE\ (\%) \leqslant 125e^{-2(Co+Cm)/3} + 75$$

where Co and Cm represent the observed and predicted PM concentrations, respectively.

We added above criteria equations in the Supplemental Materials.

2. Emission inventory is a key part in air quality modeling, but the authors do not write much about emission inventory (based on which year, what the sectors are, and etc.). Besides, the simulation period is as long as 9 years. It is better if the authors can consider the changes in emissions over this period.

Responses: The details of emission inventory were provided in the Part I paper, so we didn't repeat in this paper. This is clarified on lines 144-146 of the revised manuscript. A few additional details are provided below to help the reviewer understand what was done.

The emission inventory base year was 2000. We did consider the changes in emissions over the modeling period when information is available. The EMFAC 2007 model (CARB 2008) was used to scale the mobile emissions using predicted temperature and relative humidity fields through the entire nine-year modeling episode. Biogenic emissions were generated using the Biogenic Emissions Inventory System v3.14 (BEIS3.14), which includes a 1-km resolution land cover database with

230 different vegetation types (Vukovich and Pierce 2002). Sea-salt emissions were generated on-line based on the formulation described by de Leeuw et al. (de Leeuw et al. 2000) for the surf zone and the formulation described by Gong (Gong 2003) for the open ocean. Emissions from wildfires and open burning at 1 km × 1 km resolution were obtained from the Fire INventory from NCAR (FINN) (Hodzic et al. 2007; Wiedinmyer et al. 2011). These emissions included the changes over the model period. However, for other emissions, for example, stationary sources and residential wood burning emissions were kept unchanged due to unavailable information to scale the emissions.

3. The authors compared ratio from model to CMB derived ratio. Are the CMB derived ratios 100% reliable? How is its uncertainty? And it is also better to introduce the CMB method in the manuscript.
Responses: The uncertainties of CMB derived SOA range from 1% - 22% (Daher et al. 2012). The information was added in the caption of Figure 3.

We used the CMB results to evaluate the model performance of SOA and POA predicted by the UCD/CIT model. Because the CMB method is not the focus of this study, we think it is not necessary to introduce the CMB method in the manuscript. The references on the CMB method were provided in the original manuscript.

4. The authors discuss the impact of vapor wall losses on SOA concentrations. How will it affect model evaluation? In Figure 1, model overpredicts OA at some sites. After vapor wall losses correction, model may mismatch with observations.
Responses: The vapor wall losses correction didn't significantly affect the model evaluation results for total OA. The predicted SOA/TOA fractions in the base model is very low (generally < 10%) at the observation sites located in urban areas (Figure 4 and Figure S1). Even though the vapor wall loss correction led to substantial increase of SOA concentrations, the total OA concentrations didn't increase much.

A sentence was added in the manuscript in the lines 363-366:
"Due to low SOA/TOA fractions (< 10%) at the observation sites located in urban areas (Figure 4 and Figure S1), the substantial increase of SOA by the vapor wall loss corrections does not strongly change the total OA concentrations and therefore does not significantly affect the model evaluation results shown in Figure 1."

References:
CARB, 2008: Calculating emission inventories for vehicles in California. *User's Guide EMFAC 2007 version 2.30*, **Accessed in 2010**.
Daher, N., and Coauthors, 2012: Characterization, sources and redox activity of fine and coarse particulate matter in Milan, Italy. *Atmos Environ*, **49,** 130-141.
de Leeuw, G., F. P. Neele, M. Hill, M. H. Smith, and E. Vignali, 2000: Production of sea spray aerosol in the surf zone. *J Geophys Res-Atmos*, **105,** 29397-29409.
Gong, S. L., 2003: A parameterization of sea-salt aerosol source function for sub- and super-micron

particles. *Global Biogeochemical Cycles*, **17**.

Hodzic, A., S. Madronich, B. Bohn, S. Massie, L. Menut, and C. Wiedinmyer, 2007: Wildfire particulate matter in Europe during summer 2003: meso-scale modeling of smoke emissions, transport and radiative effects. *Atmos Chem Phys*, **7,** 4043-4064.

Vukovich, J. M., and T. Pierce, 2002: The Implementation of BEIS3 within the SMOKE modeling framework. MCNC-Environmental Modeling Center, Research Triangle Park and National Oceanic and Atmospheric Administration.

Wiedinmyer, C., S. K. Akagi, R. J. Yokelson, L. K. Emmons, J. A. Al-Saadi, J. J. Orlando, and A. J. Soja, 2011: The Fire INventory from NCAR (FINN): a high resolution global model to estimate the emissions from open burning. *Geosci Model Dev*, **4,** 625-641.

---

## Author Comment (AC2) · 15 Mar 2017

Dear Reviewer,

Thank you for the comments to help improve the quality of the paper. We have revised the manuscript to address your comments and a detailed response to each comment is provided in this file.

The comments are in regular font, the responses are in red, and the changes in the manuscript are in blue.

**Anonymous Referee #2**

The authors provided a long-term analysis for the spatial distribution of PM0.1, and its components including POA and SOA. By using the source apportionment method, the authors further discussed the contribution of different sources on PM0.1 and its components. The article is generally well-written, and has clearly expressed the conclusions clearly by showing convincing data analysis. I will suggest the paper published on ACP, after the authors address my following suggestions:

Pg 10: define the metric used for the evaluation: MFB and MFE. Can put the equation into the Supporting.

Responses: Accepted. We put the definitions in the Supplemental Materials and referred to the definition in the main text.

Pg 10: "in the first paper in the series": if the authors claimed this paper as "the fourth in the series" (Pg 5 line 70), then I suggest the authors change "the first paper" to "the third : : :." to avoid confusion, or do the other way.

Responses: We changed to "Part I paper" to avoid confusion.

Pg 11, line 189-190: I assume the authors were still talking about the winter when they say "Wood smoke is predicted to be : : :."?

Responses: To be more accurate, we changed to "Wood smoke is predicted to be the dominant OC source in winter….".

Pg 11, line 192-193: the authors implied that the overestimation of the PM2.5 in the San Jose site was due to the overestimated emission inventory. So how did the authors make that conclusion? Was the emission inventory data significant different from the other places, or more uncertain compared with others?

Responses: The PM2.5 OC was consistently over-predicted at San Jose in all winter seasons during the 9 year period (Figure 1b). While at other sites, wintertime PM2.5 OC was not always over-predicted in all winters. Considering that wood burning is the dominant source of winter PM2.5 OC in California, and similar meteorological performance among these sites, we attributed the consistent over-prediction of PM2.5 OC at San Jose more likely to overestimated wood burning emissions at this location.

Surveys of home heating methods conducted by the Bay Area Air Quality

Management District (BAAQMD) found that wood smoke emissions inventories were over-estimated in San Jose for the years 2012 and 2013. While these years are outside of the analysis window in the current manuscript, these findings support the hypothesis that wood smoke emissions in San Jose are over-estimated in the years 2000-2009.

This has been explained on lines 200-202 of the revised manuscript.

Pg 12: line 217-219: I suggest the authors move the brief introduction of the 6 Obs sites into Pg 10 to Pg 10 in front of Fig. 1. Also can the authors comment why they didn't use the EI Cajon site to evaluate the model's performance of simulating in PM2.5 in Fig. 1?
Responses: We moved the brief introduction of the observation sties to Pg10 in front of Fig.1. We kept six sites in Fig. 1 to make the figure clearer, but in the revised manuscript, we added El Cajon in Fig. 1.

Pg 15, line 287: change "PM2.5" to "PM2.5-SOA fraction" or "that in PM2.5". Also the authors concluded that the SOA fraction in PM0.1 lower than that in PM2.5, but in Figure 4, we can see the fractions are higher in PM0.1 than PM2.5 in rural areas. Can the authors explain why?
Responses: We changed to "that in PM2.5" and we added "in urban areas" to be more accurate.

The SOA/TOA fractions in PM0.1 are generally low at all locations where primary combustion emissions are significant. This includes all major urban areas or locations with major transportation corridors. The PM0.1 SOA/TOA fraction increases in regions with very low primary combustion emissions because few natural sources emit primary OA in this size range. Natural sources including windblown dust contribute more to the PM2.5 size fraction than the PM0.1 size fraction in these remote regions, which explains the different behavior illustrated in Figure 4 and Figure S1. All concentrations are very low in these remote regions, and so the points have minor importance for health effects analysis. We feel that the extended discussion would risk confusing the reader, and so we have made these points in the Figure caption for S1 rather than in the main text.

Pg 34, in Figure 7 and others, also in the supplementary, I am confused about the meaning of colorbar. I thought it stands for the fractions from each source category in the total PM0.1 POA, but it seems not. What is the "maximum concentration value", maximum of the monthly mean or maximum of the yearly mean? Also how the authors made the conclusion that the dominant regional sources are "wood smoke, meat cooking : : :"? Looking at the map, most of the data are in the range of "0-10" %, and you can't tell which regions are in the 1% and which regions are in the 9%. For sources with a Max value of 900 but fractions around 1% may not be larger than the

source with a Max value of 120 and fractions around 9%. Please quantify the fractions from each source before making conclusion. Also consider doing this for other similar plots.

Responses: The maximum concentration value is the maximum 9-year average concentrations. We made the changes in the figure captions to be clearer.

The dominant sources were determined based on the total contributions of the sources region wide. We added the fraction values in the discussion and we did the same changes for the discussion of Figure 8.

Pg 43 & 44: Switch the order S4 and S5 to follow when they are mentioned in the paper.

Responses: The order of figures in the supplemental materials was changed to follow when they are mentioned in the paper.

---

## Author Comment (AC3) · 15 Mar 2017

Dear Reviewer,

Thank you for the comments to help improve the quality of the paper. We have revised the manuscript to address your comments and a detailed response to each comment is provided in this file.

The comments are in regular font, the responses are in red, and the changes in the manuscript are in blue.

**Anonymous Referee #1**

The authors discussed the concentrations and sources of primary and secondary organic aerosols in PM0.1 over California for 2000-2008 using the source-oriented UCD/CIT model. The article is overall well-written. I will suggest the paper accepted by ACP after the authors address my following questions/suggestions:

1. SOA module

The SOA module used in this study is based on the two-product method. Different SOA formation treatments could result in different results. It would be meaningful if an alternate SOA module (e.g., VBS) is applied in the future study of POA and SOA.

Responses: Thanks for the comment. Atmospheric SOA formation pathways and processes are the focus of intense research which leads to continuous evolution in our understanding about accurate SOA modeling approaches. Part of our research team has recently developed a new statistical oxidation model to simulate SOA (Cappa et al. 2013; Cappa et al. 2016; Jathar et al. 2016) that is able to study the effects of multi-generational chemistry, evaporation of SOA fragments, wall loss effects, etc. Many of these issues are also the focus of VBS modeling efforts, and so we feel that we are capturing the essence of the scientific questions even if we have not directly applied the VBS model itself. The results of the statistical oxidation modeling studies are described in Section 3.3 of the manuscript. Future applications of long-term modeling in California will improve on the 2-product model to capture the latest scientific findings, but this issue is beyond the scope of the current paper. As a result, no changes were made in the current manuscript based on this comment.

2. OC/Mass ratios

The authors discussed the underpredictions of OC/Mass ratios shown in Figure 2, which could be due to the overestimation of dust emissions. Are dust emissions affected by wind speed from WRF? Did the authors evaluate the meteorology provided by WRF? What about seasalt, since some sites are along the coast? Although most of seasalt are coarse particles, they may contribute a little bit to PM2.5?

Responses: In reality the dust emissions are affected by the wind speed and soil moisture. However, the dust emissions in our study were developed by California Air Resources Board based on average wind speeds. Therefore the WRF wind speed was not used in the dust emissions. This point has been clarified on lines 234-236 of the revised manuscript.

The WRF predictions have been evaluated against meteorological observations. The results were described in the Part I paper. We have clarified the sentence on lines 164-165 of the revised manuscript to clarify this point.

The seasalt emissions were included in the simulations. The seasalt emissions were calculated online using the WRF wind speed. The detailed description of the emissions was also provided in the Part I paper. This point has been clarified on lines 154-155 of the revised manuscript.

Some other comments:
1. Page 6, line 78, UCD/CIT has been defined in page 4 line 52
Responses: Corrected.
2. Page 8, line 136-137, do you mean BENZ (i.e., ABNZ1_X1, ABNZ1_X2, ABNZ2_X1, and ABNZ2_X2)?
Responses: Yes, we corrected the sentence.
3. Page 9, line 155, change "meteorology fields" to "meteorological fields"
Responses: Changed.
4. Page 13, line 246, "Condensation of SOA", do you mean the condensation of volatile VOCs?
Responses: We changed the sentence to "Condensation of the semi-volatile products to form SOA".
5. Page 15, line 277, "some important sources", could you please provide some specific sources that for Riverside case?
Responses: We speculate the missing sources are mostly likely some area sources, such as residential and/or agricultural waste emissions. With no solid evidence, we are not sure what exactly the sources are, so we don't want to give specific names in the manuscript to avoid providing misleading information to readers. No changes were made for this comment.
6. Page 15, line 287, do you mean less POA converted to SOA in ultrafine size range?
Responses: No, we mean PM0.1 OA is more of POA and less of SOA, compared to PM2.5 OA. The sentence has been clarified in the revised manuscript on line 295.
7. Page 16, line 311, You may want to switch the order of supplementary figures.
Responses: Accepted. The order of figures in the supplemental materials was changed to follow when they are mentioned in the paper.

8. Page 31, Figure 4b, why are PM0.1 SOA/TOA ratios very high over the southeastern corner? Is that partly due to boundary conditions?
Responses: That is because very low POA concentrations (as can be seen in Figure 7), i.e., very low anthropogenic emissions, in that region. No changes were made for this comment.

References:
Cappa, C. D., X. Zhang, C. L. Loza, J. S. Craven, L. D. Yee, and J. H. Seinfeld, 2013: Application of the Statistical Oxidation Model (SOM) to Secondary Organic Aerosol formation from photooxidation of

C-12 alkanes. *Atmos Chem Phys*, **13,** 1591-1606.

Cappa, C. D., S. H. Jathar, M. J. Kleeman, K. S. Docherty, J. L. Jimenez, J. H. Seinfeld, and A. S. Wexler, 2016: Simulating secondary organic aerosol in a regional air quality model using the statistical oxidation model - Part 2: Assessing the influence of vapor wall losses. *Atmos Chem Phys*, **16,** 3041-3059.

Jathar, S. H., C. D. Cappa, A. S. Wexler, J. H. Seinfeld, and M. J. Kleeman, 2016: Simulating secondary organic aerosol in a regional air quality model using the statistical oxidation model - Part 1: Assessing the influence of constrained multi-generational ageing. *Atmos Chem Phys*, **16,** 2309-2322.